# Diffeomorphic Template Transformers

## Abstract

In this paper we propose a spatial transformer network where the spatial transformations are limited to the group of diffeomorphisms. Diffeomorphic transformations are a kind of homeomorphism, which by definition preserve topology, a compelling property in certain applications. We apply this diffemorphic spatial transformer to model the output of a neural network as a topology preserving mapping of a prior shape. By carefully choosing the prior shape we can enforce properties on the output of the network without requiring any changes to the loss function, such as smooth boundaries and a hard constraint on the number of connected components. The diffeomorphic transformer networks outperform their non-diffeomorphic precursors when applied to learn data invariances in classification tasks. On a breast tissue segmentation task, we show that the approach is robust and flexible enough to deform simple artificial priors, such as Gaussian-shaped prior energies, into high-quality predictive probability densities. In addition to desirable topological properties, the segmentation maps have competitive quantitative fidelity compared to those obtained by direct estimation (i.e. plain U-Net).

## 1 Introduction

The success of Convolutional Neural Networks (CNNs) in many modeling tasks is often attributed to their depth and inductive bias. An important inductive bias in CNNs is spatial symmetry (e.g. translational equivariance) which are embedded in the architecture through weight-sharing constraints. Alternatively, spatial transformers constrain networks through predicted spatial affine or thin-plate-spline transformations. In this work, we investigate a special type of spatial transformer, where the transformations are limited to flexible diffeomorphisms. Diffeomorphisms belong to the group of homeomorphisms that preserve topology by design, and thereby guarantee that relations between structures remain, i.e. connected (sub-)regions to stay connected.

We propose to use such diffeomorphic spatial transformer in a template transformer setting (Lee et al., 2019), where a prior shape is deformed to the output of the model. Here a neural network is used to predict the deformation of the shape, rather than the output itself. By introducing a diffeomorphic mapping of a prior shape, and carefully choosing properties of the prior shape, we can enforce desirable properties on the output, such as a smooth decision boundary or a constraint on the number of connected components.

To obtain flexible diffeomorphic transformations, we use a technique known as *scaling-and-squaring* which has been successfully applied in the context of image registration in prior work (Dalca et al., 2018), but has received relatively little attention in other areas in machine learning. In an attempt to increase flexibility of the flow, we try to approximate a time-dependent parameterisation using Baker-Campbell-Hausdorff (BCH) formula, rather than a stationary field. Hereby, diffeomorphic constraints are directly built into the architecture itself, not requiring any changes to the loss function.

Experimentally, we first validate the diffeomorphic spatial transformer to learn data-invariances in a MNIST handwritten digits classification task, as proposed by (Jaderberg et al., 2015) to evaluate the original spatial transformer. The results show that better results can be achieved by employing diffeomorphic transformations. Additionally, we explore the use of diffeomorphic mappings in a spatial template transformer set-up for 3D medical breast tissue segmentation. We find that the diffeomorphic spatial transformer is able to deform simple prior shapes, such as a normally distributed energy, into high-quality predictive probability densities. We are successful in limiting the number of connected components in the output and achieve competitive performance measured by quantitative metrics compared to direct estimation of class probabilities.

## 2 RELATED WORK

Spatial Transformers were introduced by Jaderberg et al. (2015) as a learnable module that deform an input image, and can be incorporated into CNNs for various tasks. In Spatial Transformer Networks (STNs), the module is used to learn data invariances in order to do better in image classification tasks. The work focuses on simple linear transformations (e.g. translations, rotations, affine) but also allows for more flexible mappings such as thin plate spline (TPS) transformations. The use of spatial transformations in template transformer setting was first proposed by Lee et al. (2019), but does not use diffeomorphisms and requires defining a discrete image as shape prior.

In the field of image registration, diffeomorphisms have been actively studied and have been successfully applied in a variety of methods including LDDMM by Beg et al. (2005), Diffeomorphic Demons by Vercauteren et al. (2009), and SyN by Avants et al. (2008). More recently, efforts have been made to fuse such diffeomorphic image registration approaches with neural networks (Dalca et al. (2018), Haskins et al. (2020)). It is well known that although these models mathematically describe diffeomorphisms, transformations are not always diffeomorphic; in practice and negative Jacobian determinants can still occur due to approximation errors. To reduce such errors, additional regularisation is often applied (Bro-Nielsen and Gramkow (1996), Ashburner (2007), Dalca et al. (2018)), but typically requries careful tuning.

Image registration has also been applied to perform segmentation by deforming a basis template commonly referred to as an 'atlas' onto a target image (Rohlfing et al. (2005), Fortunati et al. (2013)), for instance by combining (e.g. averaging) manually labelled training annotations (Gee et al., 1993).

There have been some studies that investigated how to obtain smoother segmentation boundaries in neural-based image registration. For instance, Monteiro et al. (2020) proposed to model spatial correlation by modeling joint distributions over entire label maps, in contrast to pixel-wise estimates. In other work, post-processing steps have been applied in order to smooth predictions or to enforce topological constraints (Chlebus et al. (2018), Jafari et al. (2016)).

There have been some studies that try to enforce more consistent topology during training of neural network, but often use a soft constraint that required alteration of the loss function, such as in Hu et al. (2019), and GAN-based approaches which in addition require a separately trained discriminator model Sekuboyina et al. (2018).

Lastly, there have been some studies in which diffeomorphisms in context of spatial transformer networks were investigated. In Skafte Detlefsen et al. (2018), subsequent layers of spatial transformer layers with piece-wise affine transformations (PCAB) were used to construct a diffeomorphic neural network, but requires a tessellation strategy (Freifeld et al. (2015), Freifeld et al. (2017)). In Deep Diffeomorphic Normalizing Flows (Salman et al. (2018)) a neural network is used to predict diffeomorphic transformations as normalizing flow but to obtain more expressive posteriors for variational inference.

## 3 DIFFEOMORPHIC SPATIAL TRANSFORMERS

The Spatial Transformer is a learnable module which explicitly allows for spatial manipulation of data within a neural network. The module takes an input feature map $U$ passed through a learnable function which regresses the transformation parameters $\theta$. A spatial grid $G$ over the output is transformed to an output grid $\mathcal{T}_\theta(G)$, which is applied to the input $U$ to produce the output $O$. In the original spatial transformer, $\theta$ could represent arbitrary parameterised mappings such as a simple rotation, translation or affine transformation matrices. We propose flexible transformations in the group of diffeomorphisms $\mathcal{T}_\theta \in \mathcal{D}$, which preserve topology, by continuity and continuity of the inverse.

In Section 4, we will describe how we can use a diffeomorphic spatial transformer to warp a shape prior, as illustrated in Figure 1, in a template transformer setting illustrated in Figure 2.

**Diffeomorphic Transformation**    Let us define the diffeomorphic mapping $\phi = \psi_v^{(1)} \in \mathcal{D}$ using an ordinary differential equation (ODE):

$$\frac{\partial \psi_v^{(t)}(\boldsymbol{x})}{\partial t} = v(\psi_v^{(t)}(\boldsymbol{x})) \tag{1}$$

where $v$ is a stationary velocity field, $\psi_v^{(0)} = Id$ is the identity transformation and $t$ is time. By integrating over unit time we obtain $\psi_v^{(1)}$, the time 1 flow of the stationary velocity field $v$.

The most basic way to solve an ordinary differential equation from some initial point $\boldsymbol{x}_0$ is Euler's method, in which the trajectory is approximated by taking small discrete steps and adding the difference to the running approximation in time. The method is straightforward to implement, but may take many steps to converge to good approximations. In this work, we will use a technique known as *scaling-and-squaring* (Moler and Van Loan, 2003), which allows for fast exponentiation of stationary velocity fields and thus the solution to the ODE defined in Equation 1.

**Scaling-and-Squaring**    To solve the ODE from Equation 1, with a stationary velocity field $v$ and the solution is the matrix exponential $\phi = \exp(v)$, we use is the scaling-and-squaring method (Moler and Van Loan (2003), Arsigny et al. (2006)). The method is very similar to Euler's method, but is typically more efficient by exploiting the relation $\exp(v) = \exp(v/2^T)^{2^T}$ with $T \in \mathbb{N}$ together with the fact that $\exp(v)$ can be well approximated by a Padé or Taylor approximation near the origin (i.e. for small $||v||$). The main idea is to pick a certain step size $T$ such that $||v||/2^T < 0.5$ and divide the diagonal values in $v$ by the power integral $2^T$ to obtain the approximation for $\exp(v/2^T) \approx Id + v/2^T$ and then squaring (self-composing) it $T$ times to find obtain approximate solution for $\exp(v)$.

---

**Algorithm 1:** Approximating $\phi = \exp(v)$ using scaling-and-squaring

---

**Result:** $\phi = \exp(v)$
$T \leftarrow \text{ceil}(\log_2(\max(||v||) + 1)$
$\phi_0 \leftarrow v/2^T$
**for** $t = 1$ to $T$ **do**
$\quad | \quad \phi_t \leftarrow \phi_{t-1} \circ \phi_{t-1}$
**end**

---

The approach can efficiently be implemented in existing numerical differentiation frameworks such as PyTorch or Tensorflow by element-wise dividing the vector components in velocity field $v$ by $2^T$ and then self-composing the resulting field $2^T$ times using the linear grid sampling operation defined in Section 3.[1]

**Spatial Sampling**    To perform a spatial transformation on the input feature map, a sampler takes a set of sampling points $\mathcal{T}_\theta(G)$, along with an input feature map $U = I$ with input image $I$ to produce output $O$. In case of template transformer, explained in Section 4, the input feature map would be a concatenation $U = I \circ S$ of an input image with some prior shape $S$.

We follow the general sampling framework described in Jaderberg et al. (2015), defined for arbitrary sampling kernels of which the (sub)gradients can be computed, and the 3D trilinear interpolation in particular:

$$O_i^c = \sum_h^H \sum_w^W \sum_d^D I_{hwd}^c \max(0, 1 - |y_i^c - h|) \max(0, 1 - |x_i^c - w|) \max(0, 1 - |z_i^c - d|) \tag{2}$$

This procedure should be differentiable with respect to both the sampling grid coordinates and the input feature map by using partial (sub)gradients, allowing it to be used in conjunction with backpropagation.

---

[1]For our experiments, we utilized the `F.grid_sample` function in PyTorch 1.6 to perform grid sampling.

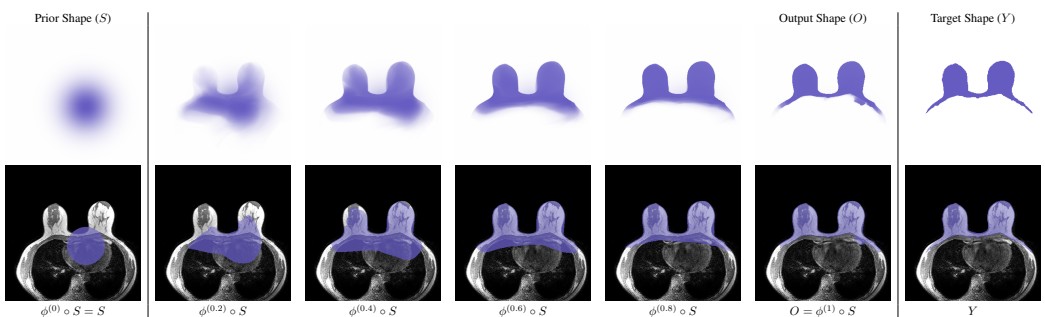

Figure 1: Illustration of diffeomorphism integrated over in time applied to shape prior. RIGHT: Target annotation. TOP: Class probabilities over voxel location. BOTTOM: Thresholded class probabilities ($p > 0.5$) laid over the input.

**Baker–Campbell–Hausdorff formula**   Instead of parameterising our flow by a single stationary velocity field, we might also think of a piece-wise time-dependent sequence of vector fields. By parameterising the deformation as a time-dependent sequence of velocities we hope improve predictive performance by sequentially modeling larger movements first and detailed refinements thereafter. Composing multiple diffeomorphic transformations will also yield a diffeomorphic transformation, as the space of diffeomorphic transformations $\mathcal{D}$ is an algebraic group that is closed under the composition operation. The scaling-and-squaring algorithm offers an efficient way to find diffeomorphic transformations from a stationary vector field, but can not straightforwardly be applied to such time-dependent parameterisations. To address this, we can combine two timepoints, now $A$ and $B$ for simplicity of notation, to form the Lie exponential mapping:

$$\exp(Z) = \exp(A)\exp(B) \tag{3}$$

and apply the Baker-Campbell-Hausdorff (BCH) formula up to a certain order to approximate

$$Z = \text{bch}(A, B) = \sum_{n=1}^{\infty} z_n(A, B) = A + B + \frac{1}{2}[A, B] + \frac{1}{12}[A, [A, B]] - \frac{1}{12}[B, [A, B]] + \cdots \tag{4}$$

where $[\cdot, \cdot]$ is the Lie bracket. We apply the formula to approximate the logarithm of matrix exponentials of two noncommutative velocity fields $Z = \log(\exp(A)\exp(B))$ and then use scaling-and-squaring one time to find the exponential $\exp(Z)$.

**Binary Tree Composition**   Naive composition of the $T$ diffeomorphic transformations would result into a long chain of composition operations $\Phi = (((((( \phi_1 \circ \phi_2) \circ \phi_3) \circ \phi_4) \circ \phi_5) \cdots) \cdots \circ \phi_{T-1}) \circ \phi_T)$. To reduce possible interpolation errors in the resampling from growing as a result of such repetitive composing, we compose the field using a binary tree scheme $\Phi = (((\phi_1 \circ \phi_2) \circ (\phi_3 \circ \phi_4)) \circ (\cdots \circ (\phi_{T-1} \circ \phi_T)))$. Treating the composition scheme as a tree structure, the depth now scales in an order of complexity $\mathcal{O}(T)$ compared to $\mathcal{O}(\log(T))$ when using naive composition, reducing the maximum number of times an BCH approximation is repetitively applied to a single timepoint.

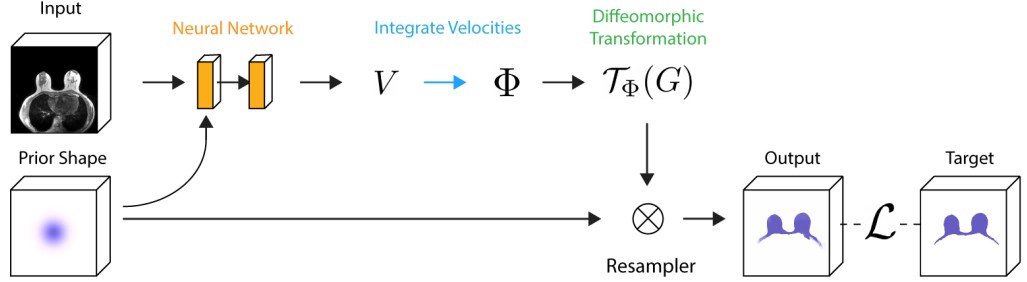

Figure 2: Illustration of Diffeomorphic Spatial Template Transformer: a neural network predicts a set of velocity fields $V = f(I, S)$ from an input image $I$ and prior shape $S$. The fields are integrated to a diffeomorphic transformation grid $\mathcal{T}_\Phi(G)$, which transforms the prior shape into an output $O$, while preserving topology. The model is trained end-to-end so that output $O$ matches target $Y$.

## 4 DIFFEOMORPHIC TEMPLATE TRANSFORMER

Now that we have defined how to obtain a flexible diffeomorphic spatial transformer, we will investigate its use in a template transformer setting. We define the output of a segmentation model as diffeomorphic transformation of a prior shape, based on input image and the prior shape. By carefully choosing the prior shape and its properties, we obtain explicit control over the properties of the output, such as the number of connected components.

Let the input of the model $f$ be a feature map $U \in \mathbb{R}^{H \times W \times D \times 2C} = I \circ S$ be a concatenation of an input image $I \in \mathbb{R}^{H \times W \times D \times C}$ and an prior shape $S \in \mathbb{R}^{H \times W \times D \times C}$ along the channel dimension, with width $W$, height $H$, depth $D$ and channels $C$ that outputs a set of $T$ velocity fields $V = f(U) = \{v_t\}_{t=1}^T$, where the fields $v_t \in \mathbb{R}^{H \times W \times D \times 3}$ are concatenated along the channel dimension of the output. Then we compute the diffeomorphic transformation $\Phi = \prod_{t=1}^T \exp(v_t)$ by approximating the product of matrix exponentials as discussed in Section 3. Lastly, we subsample the pixels in the original prior shape $S$ using the diffeomorphic grid to obtain the output of the model $O = \mathcal{T}_\Phi(G)(S)$ as explained in Section 3. The resulting model is illustrated in Figure 2.

### 4.1 PRIOR SHAPE

The template transformer can in principle use any prior shape, such as a discrete image by averaging annotations (or 'atlas' (Gee et al., 1993) (Cabezas et al., 2011)). But, by carefully choosing a prior shape and by continuity of the diffeomorphic transformation, we can enforce properties such as single connected component and smooth boundaries on the model output. In this paper we aim to keep the prior shape a simple more general form to emphasise the expressivity of the diffeomorphism in our experiments. We choose an analytical shape prior inspired by the generalised multivariate Gaussian, and define the probability of a voxel at location $\boldsymbol{x}$ belonging to the main class by

$$p(\boldsymbol{x}; \boldsymbol{\mu}, \boldsymbol{\Sigma}, \beta) = \exp\left[-((\boldsymbol{x} - \boldsymbol{\mu})^T \boldsymbol{\Sigma}^{-1}(\boldsymbol{x} - \boldsymbol{\mu}))^\beta \cdot \log(2)\right] \tag{5}$$

where $\boldsymbol{\mu}$, $\boldsymbol{\Sigma}$ and $\beta$ directly influence the mean, (co)variances and kurtosis of the prior shape in the spatial domain, and can be kept fixed or trained as part of the model parameters (see Section 5.4). The $\log(2)$ factor ensures that the decision boundary (p=0.5) is independent from $\beta$.

## 5 EXPERIMENTS AND RESULTS

The diffeomorphic spatial transformer is evaluated on two tasks: a classification task using handwritten MNIST dataset and a medical 3D breast tissue segmentation problem in the template transformer setting. In both settings, its performance is compared with its non-diffeomorphic counterparts. For the segmentation, we additionally analyse the effect of training different shape prior parameters in Section 5.4.

Table 1: Quantitative evaluation of diffeomorphic spatial transformer on MNIST classification task when learning to augment the input with various types of Spatial Transformer Networks (STNs).

| Model | Type | Parameter Count | Accuracy | $\% |J_\phi| < 0$ |
|---|---|---|---|---|
| CNN (baseline) | | 27100 | $96.21 \pm 0.39$ | - |
| CNN + TPS-STN (Jaderberg et al. (2015)) | $2 \times 2$-grid | $27100 + 26998$ | $96.60 \pm 0.46$ | $0.03 \pm 0.007$ |
| CNN + TPS-STN | $3 \times 3$-grid | $27100 + 27508$ | $95.92 \pm 0.40$ | $0.03 \pm 0.003$ |
| CNN + TPS-STN | $5 \times 5$-grid | $27100 + 29140$ | $96.63 \pm 0.32$ | $0.04 \pm 0.002$ |
| CNN + TPS-STN | $12 \times 12$-grid | $27100 + 41278$ | $96.49 \pm 0.29$ | $0.05 \pm 0.006$ |
| CNN + TPS-STN | $16 \times 16$-grid | $27100 + 52702$ | $96.12 \pm 0.43$ | $0.07 \pm 0.010$ |
| CNN + Diffeomorphic-STN + $||\nabla\phi||_2$ penalty (**Ours**) | $28 \times 28$-grid | $27100 + 29552$ | $96.45 \pm 0.48$ | $0.11 \pm 0.009$ |
| CNN + Non-diffeomorphic-STN (w/o field integration) | $28 \times 28$-grid | $27100 + 29552$ | $97.30 \pm 0.52$ | $0.28 \pm 0.031$ |
| CNN + Diffeomorphic-STN (**Ours**) | $28 \times 28$-grid | $27100 + 29552$ | $\mathbf{97.34 \pm 0.55}$ | $0.22 \pm 0.038$ |

## 5.1 IMPLEMENTATION

For the MNIST classification experiments, we adapt an existing spatial transformer network implementation on Github[2] and added transformations using diffeomorphic vector fields. We train for 10 epochs with a batch size of 64 and a learning rate of 0.001 ($\beta_1 = 0.9, \beta_2 = 0.999$) without any further learning rate decay.

For the 3D breast tissue experiments, we trained for 40k iterations using Adam (Kingma and Ba, 2014) with a batch size of 1 and a learning rate of 0.0002 ($\beta_1 = 0.9, \beta_2 = 0.999$) decayed with cosine annealing (Loshchilov and Hutter, 2016). The input was normalised using 1-99 percentile normalisation (Patrice et al., 2018) and training samples consist of randomly sampled $128 \times 128 \times 64$ patches. For the neural network, a 3D U-Net (Ronneberger et al., 2015), 4 times spatial down- and up-sampling using linear interpolation (Odena et al., 2016), instance normalisation (Ulyanov et al., 2016) and the Leaky ReLU (slope 0.2) activation functions. Lastly, an hyperbolic tangent scaled with $\alpha = 256$ limits the vector components of the vector fields to the range $[-\alpha, \alpha]$. The network uses a single input channel and $3 \times T$ output channels, with $T = 4$ and a 2nd order BCH approximation. We train the model using a standard cross entropy loss.

The prior shape was initialized as a centred Gaussian with diagonal co-variance matrix with diagonal components set at half the volume size of the corresponding dimension. Aside from trying to optimise these values as part of the model parameters (see Section 5.4), we did not perform hyper-parameter tuning on these values. All experiments were performed on a single NVIDIA GeForce RTX 2080 Ti.

## 5.2 MNIST EXPERIMENTS

In this experiment, analogous to the one described in Jaderberg et al. (2015), we take a simple CNN classifier with and without an additional spatial transformers added to the beginning of the network and train it end-to-end. We train and evaluate the model on the well-known MNIST dataset (LeCun et al., 2010) comprising 60000 training and 10000 testing images with size of $28 \times 28$ pixels that contain hand-drawn numbers in the range 0 to 10, with images randomly rotated to an angle $\alpha$ uniformly sampled from the range $[-90, 90]$ degrees. The idea is that the spatial transformer can learn invariances in the data (e.g. translation) and thereby aid the classifier. The spatial transformer networks were designed in such a way that they have approximately the same parameter count, and the same classifier model was used in all cases. The experiment was repeated 20 times and standard deviations were reported. For fair comparison, we did not tune hyper-parameters in favor of the diffeomorphic-STN.

In Table 1, the results between different types of spatial transformers are shown. We find that our diffeomorphic spatial transformer network results in highest predictive accuracy, compared to non-diffeomorphic and more course thin-plate-spline (TPS) spatial transformers. The TPS models generate inherently smoother fields, and are therefore less prone to folding resulting in fewer negative Jacobian determinants. On the other hand, coarser TPS grids have less flexibility, which would make them unsuitable for application in complex anatomical segmentation tasks. We do observe that integration in our diffeomorphic model helps lowering the number of negative Jacobian determinants when compared to the same model without integration. In addition, we performed an experiment with an added regularisation term to the loss penalizing the spatial gradient of the field $\lambda||\nabla\phi||_2$, where $\lambda = 10$ controls the amount of regularisation. We find that this helps to limit the number of negative Jacobian determinants, but also negatively impacts overall accuracy.

---

[2]Used TPS-STN implementation: `https://github.com/WarBean/tps_stn_pytorch`

Table 2: Quantitative evaluation of breast tissue segmentation on breast tissue segmentation dataset for different methods, indicating whether method uses a template, preserves topology by-design, and comparison of loss, Sørensen–Dice coefficient, Hausdorff distance, ratio of negative Jacobian determinants and average number of connected components on the validation set.

| Model | Template | Preserves Topology | Dice Score | Hausdorff Distance | $\% \|J_\phi\| < 0$ | Connected Components |
|---|---|---|---|---|---|---|
| U-Net (direct estimation) | | | $0.846 \pm 0.24$ | $12.62 \pm 15.48$ | - | 51.79 |
| Spatial Template Transformer ((Lee et al., 2019)) + Shape Prior (**ours**) | ✓ | | $\mathbf{0.877 \pm 0.21}$ | $12.58 \pm 12.91$ | $0.43 \pm 0.01$ | 19.68 |
| Diffeomorphic Spatial Template Transformer (**ours**) + Shape Prior (**ours**) | ✓ | ✓ | $0.803 \pm 0.298$ | $10.10 \pm 6.51$ | $0.39 \pm 0.009$ | 31.20 |
| Diffeomorphic Spatial Template Transformer (**ours**) + Trained Shape Prior (**ours**) | ✓ | ✓ | $0.822 \pm 0.249$ | $\mathbf{9.81 \pm 7.39}$ | $\mathbf{0.15 \pm 0.01}$ | **4.75** |

## 5.3 SEGMENTATION EXPERIMENTS

To assess the applicability of diffeomorphic spatial transformer in a template transformer setting, we compare our differentiable spatial transformer with and without trained shape prior with direct estimation (i.e. plain U-Net) on a breast tissue segmentation task. We also evaluate a non-diffeomorphic spatial transformer, as is done in Lee et al. (2019), but applied it in combination with our shape prior as it was not obvious to us how to create a discrete 3D template from our 2D annotations.

The dataset comprises 20 training volumes and 20 evaluation volumes of dynamic contrast enhancement series of subjects with extremely dense breast tissue (Volpara Density Grade 4). Each series contains DCE-MRI images ($384 \times 384 \times 60$ voxels with spacing $0.97 \times 0.97 \times 3.00$ mm resampled to 2.5mm$^3$) acquired on a 3.0T Achieva or Ingenia Philips system in the axial plane with bilateral anatomic coverage. A randomly selected axial 2D slice was annotated to be used for training and evaluation labels. All annotations make up a single connected component.

Performance was measured using the well-known Sørensen–Dice coefficient ($F_1$-score) (Dice (1945), Sorensen (1948)) and Hausdorff distance (Hausdorff, 1978) metrics. In addition, we measure the percentage of negative Jacobian determinants of the approximated flow, a well-known metric for deformation quality in image registration that measures the amount of folding. Lastly, we evaluate whether the connected components in the thresholded output ($p > 0.5$) is close to 1, as should be the case without approximation errors. The HD and CC metrics on this medical imaging tasks are particularly important indicating high-quality and robust results.

In Table 2, a comparison of a spatial template transformer with fixed shape prior, a diffeomorphic spatial transformer with fixed shape prior and a diffeomorphic spatial transformer with trained shape prior (trained mean $\boldsymbol{\mu}$, diagonal covariance diag($\boldsymbol{\Sigma}$) and $\beta$) with direct estimation (i.e. plain U-Net) is shown. We find that all template transformer models perform better in terms of Hausdorff distance. The diffeomorphic spatial template transformers perform worse in terms of Sørensen–Dice coefficient, but in combination with a trained shape prior are able to reduce the number of connected components and Hausdorff distance. Lastly, we observe negative Jacobian determinants as a result of approximation errors in all template transformers, but to a lower degree in the diffeomorphic models.

## 5.4 ANALYSIS OF PRIOR SHAPES

In this section we empirically assess the impact of different prior shapes, with fixed or varying $\boldsymbol{\mu}$, $\boldsymbol{\Sigma}$ and $\beta$ parameters. In Table 3, Sørensen–Dice coefficient, Hausdorff distance, ratio of negative Jacobians $\% \|J_\phi\| < 0$ and average number of connected components are reported for different combinations of trained prior shape parameters.

Table 3: Ablation study where different parameters for the prior shapes are kept fixed or learnt.

| Trained prior shape | Dice Score | Hausdorff Distance | $\% \|J_\phi\| < 0$ | Connected Components |
|---|---|---|---|---|
| Fixed Prior Shape | $0.803 \pm 0.298$ | $10.10 \pm 6.51$ | $0.39 \pm 0.009$ | 31.20 |
| Train mean $\boldsymbol{\mu}$ | $0.820 \pm 0.262$ | $11.30 \pm 10.65$ | $0.22 \pm 0.013$ | $\mathbf{4.38 \pm 4.87}$ |
| Train scale $\sigma^2$ ($\boldsymbol{\Sigma} = \sigma^2 \boldsymbol{I}$) | $0.807 \pm 0.297$ | $9.92 \pm 12.17$ | $0.36 \pm 0.023$ | $61.16 \pm 45.1$ |
| Train diag($\boldsymbol{\Sigma}$) | $0.821 \pm 0.284$ | $\mathbf{8.66 \pm 5.57}$ | $0.47 \pm 0.007$ | $27.47 \pm 11.5$ |
| Train $\boldsymbol{\mu} + \sigma^2$ | $0.821 \pm 0.252$ | $10.10 \pm 7.84$ | $0.25 \pm 0.012$ | $5.23 \pm 5.48$ |
| Train $\boldsymbol{\mu} + $ diag($\boldsymbol{\Sigma}$) | $0.790 \pm 0.268$ | $12.58 \pm 11.64$ | $0.22 \pm 0.020$ | $5.02 \pm 5.34$ |
| Train $\boldsymbol{\mu} + $ diag($\boldsymbol{\Sigma}$) $+ \beta$ | $\mathbf{0.822 \pm 0.249}$ | $9.81 \pm 7.39$ | $\mathbf{0.15 \pm 0.012}$ | $4.75 \pm 4.37$ |

We find that learning parameters of the shape prior positively contributes to performance and helps to reduce the number of negative Jacobian determinants, most notably for learnt position $\mu$. The result suggests that, in case of more complex shape priors such as a segmentation atlas, the model could benefit from deforming the prior shape with some linear transformations (e.g. translation or affine) before being warped by the diffeomorphic transformation predicted by the network.

## 6 DISCUSSION AND CONCLUSION

We have presented a special type of spatial transformer where the spatial transformations are restricted to the group of diffeomorphisms. Diffeomorphic deformations are topology preserving by continuity and continuity of their inverse, which can be a compelling property when designing deep learning architectures. We show how expressive diffeomorphic mappings can be obtained by time-dependent parameterisation of multiple vector fields utilizing the Baker-Campbell-Hausdorff formula in combination with an efficient integration method known as scaling-and-squaring. By building these constraints directly into the architecture itself, no changes to the loss function are required. In addition, we propose to use the diffeomorphic spatial transformer in a template transformer setting, constraining the output of a neural segmentation model as topology-preserving mapping of an analytical prior shape. Hereby, we show that the diffeomorphic transform enforces smooth boundaries and explicit control over the topology of the output such as its number of connected components.

The diffeomorphic spatial transformer outperforms the original spatial transformer network when used to learn data-invariances on MNIST. In a template transformer set-up, we found that a neural network predicting a diffeomorphic mapping of a prior shape offers a flexible way to insert knowledge about the structure of the output without having to alter the loss function or optimisation scheme. We were able to warp shape priors into high-quality segmentations in a medical 3D breast tissue segmentation task, resulting in lower number of connected components and obtain higher performance in terms of Hausdorff distance but lower in terms of Dice Score compared with direct estimation (i.e. plain U-Net).

To show the effectiveness of the approach, we used a general and simple Gaussian-shaped shape prior as template. Interestingly, the method is flexible enough to find diffeomorphic mappings from such simple shapes into high-quality posteriors. We expect that designing shape priors, specifically tailored to a task (e.g. an atlas or average segmentation) might achieve even better results. It would be interesting to explore applicability on more complex anatomical structures, such as in coronary artery tree segmentation (Lee et al. (2019)).

A piece-wise constant time-dependent parameterisation performed slightly better than modeling a stationary velocity field. This surprised us, because for every diffeomorphic mapping generated by a piece-wise constant time-dependent field there also exists a single (stationary) vector field that describes the same diffeomorphism. We hypothesise that directly optimising this stationary velocity field is harder and that the time-dependent parameterisation aids the optimisation process by allowing the network to model larger and more detailed deformations separately.

We used the BCH-formula to integrate a piece-wise time-dependent velocity field using the scaling-and-squaring method. It would be interesting to see how this method relates to other ODE solvers that are capable of integrating time-dependent velocity fields, such as those proposed in Chen et al. (2018). However, we were unable to use these solvers under available hardware constraints.

In some cases, negative Jacobian determinants are still present in the obtained flows. This is likely caused by spatial discretisation and interpolation during resampling operations. Future research could assess whether the approach could benefit from regularisations of the vector fields (Ashburner, 2007), inverse consistency (Christensen, 1999) or better interpolation methods.

To conclude, we show that diffeomorphic spatial transformations can successfully be applied to preserve topology in neural networks, and for template transformer networks in particular. We have provided several insights on how to incorporate diffeomorphisms in neural network architectures for classification and segmentation. We expect that these insights can aid in tailoring neural network architectures to specific structure and geometry in data.

ACKNOWLEDGEMENTS

**Anonymised**

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

## A  BAKER-CAMPBELL-HAUSDORFF FORMULA ORDERS $z_1$ TO $z_7$

If we denote the Baker-Campbell-Hausdorff formula as

$$Z(X,Y) = \log(\exp(X)\exp(Y)) = \sum_{n=1}^{\infty} z_n(X,Y)$$

we can simplify (Van-Brunt and Visser (2016)) the lower order terms $z_1$ to $z_6$ to

$$z_1(X,Y) = X + Y$$
$$z_2(X,Y) = \frac{1}{2}(XY - YX)$$
$$z_3(X,Y) = \frac{1}{12}(X^2Y - 2XYX - XY^2 + YX^2 - 2YXY + Y^2X)$$
$$z_4(X,Y) = \frac{1}{24}(X^2Y^2 - 2XYXY + 2YXYX - Y^2X^2)$$
$$z_5(X,Y) = \frac{1}{6!}(-X^4 + 6XYXYX + 2XY^3X + 2YX^3Y + 6YXYXY - Y^4X)$$
$$z_6(X,Y) = \frac{1}{2*6!}(-2X^2Y^2XY + 6XYXYXY - XY^4X + Y^4Y)$$

Higher order $z_7$ and $z_8$ terms were not used in this study, but can be found in the Supplementary Material of Van-Brunt and Visser (2016).

## B  TIMING MEASUREMENTS

A comparison of performance in terms of inference time can be found in Table 4. Average inference time was calculated over 20 full 3d volumes on the breast tissue segmentation task on the U-Net model baseline, non-diffeomorphic model (without field-integration) and diffeomorphic models with a stationary and time-dependent vector field parameterisations. Due to integration procedures, the average inference time on the diffeomorphic template transformer models is slightly slower ($\approx 10\%$) compared to the U-Net baseline, but well within practical bounds.

Table 4: Average inference time in seconds (s) measured over 20 full 3d volumes on the U-Net baseline, non-diffeomorphic template transformer, diffeomorphic template transformer with stationary field and diffeomorphic template transformer with time-dependent field.

| Method | Average inference time over 20 full 3d volumes |
| --- | --- |
| U-Net | 1.03 s |
| U-Net + non-diffeomorphic field | 1.06 s |
| U-Net + stationary diffeomorphic field | 1.17 s |
| U-Net + diffeomorphic field **(ours)** | 1.19 s |

## C    EXAMPLE VARIATIONS IN SHAPE PRIOR PARAMETERS

To illustrate how variations in the parameters $\boldsymbol{\mu}$, $\sigma$, $\boldsymbol{\Sigma}$ and $\beta$ spatially change the shape prior, we plot the probability (white: $p = 0$ and purple $p = 1$) for each voxel $p(\boldsymbol{x}; \boldsymbol{\mu}, \boldsymbol{\Sigma}/\sigma\boldsymbol{I}, \beta)$ with different parameter values both smooth (top) and thresholded (bottom):

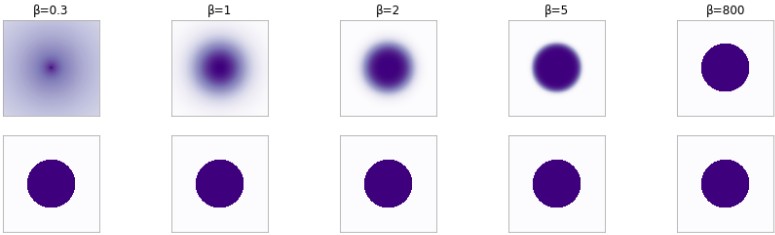

Figure 3: Shape prior under different values for $\beta$.

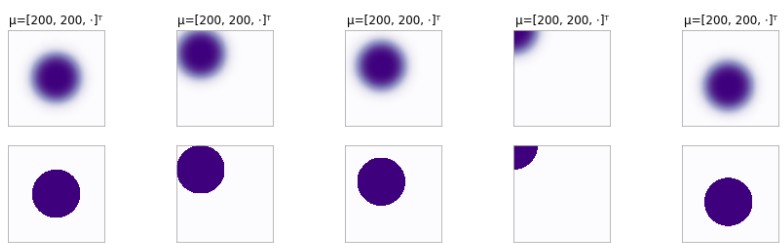

Figure 4: Shape prior under different values for $\boldsymbol{\mu}$.

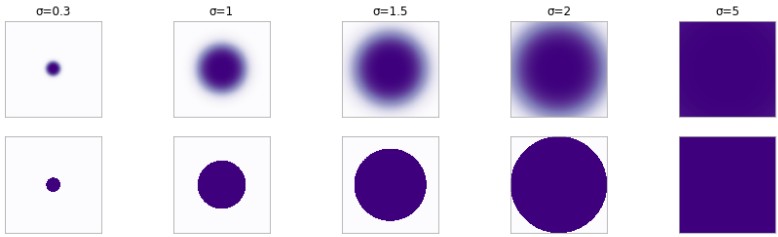

Figure 5: Shape prior under different values for $\sigma$.

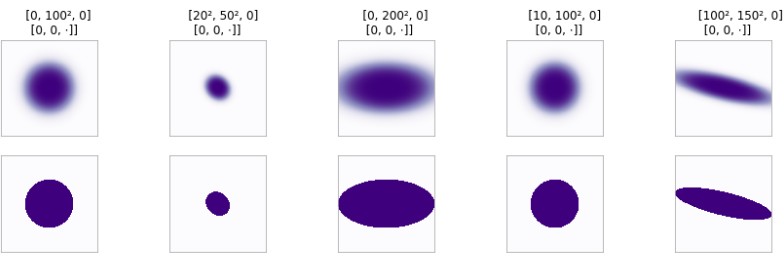

Figure 6: Shape prior under different values for $\boldsymbol{\Sigma}$.

