# OpenReview forum: "Diffeomorphic Template Transformers"
_ICLR.cc/2021/Conference — Reject_

### Official Review · AnonReviewer3 · 2020-10-28
**Review of Diffeomorphic Spatial Transformer Networks: Weak Accept**

**Rating:** 5
**Confidence:** 4

**Review:**

Summary:

The authors present a modification to spatial transformer networks that restricts the transformations to the group of diffeomorphisms. When combined with shape priors, this imposes topological constraints on the mappings produced by the network. These are important considerations in applications such as segmentation tasks where we expect there to be constraints on, for example, the number of connected components. The authors demonstrate the effectiveness of their approach in MNIST experiments and a breast tissue segmentation task.

Strengths:
1) The paper is clear and well written, and positions its contributions well in relation to existing research.

2) The method is well motivated and the procedure is reasonable. The paper tackles an important consideration when applying neural networks to segmentation tasks, as constraining to the group of diffeomorphisms imposes smoothness on the transformations that ensures the outputs share topological properties with our prior expectations (e.g. number of connected components, smoothness of boundaries).

3) The authors use the scaling-and-squaring to solve the ODEs that describe the diffeomorphic transformations, which I have not seen used in other ML studies.

Weaknesses:

1) Section 3, which describes the components of the method, can be expanded (there is space for this at the end of page 2) and reorganised to make it clearer. Specifically:

	a) the last two lines of paragraph one of page 4, starting with ‘The scaling-and-squaring...’, say this algorithm is not amenable to time-dependent parametrisations as laid out by the ODE in eq. 1. It is not immediately clear to me as to what the A and B refer to in this situation, are these the separate time steps t?

	b) I would like to see more discussion about the binary tree assumption on the composition of the diffeomorphisms. I can understand why this is a useful assumption with regards to complexity and numerical stability, but I do not have good intuition about the implications of this assumption on the transformations the model outputs.

My suspicion is the combination of these two assumptions explain some of the behaviours seen in the results.

2) The results section could also be expanded for a bit more clarity. My main concerns are around the metric that measures the percent of the Jacobian determinants that are less than zero. Is this computed from all observations or just on a hold out test set? Why is it the case that the supposedly non-diffeomorphic models in Table 1 perform better on this metric than the diffeomorphic models?

Why is there such a large discrepancy in the size of the standard deviations in Table 2?

I do not think the results are particularly convincing in terms of improvements in the quality or the accuracy of the methods compared with, but the point of the paper stands if the behaviour of this metric is properly discussed.


Reasons for score:
I vote for accepting the paper. On the whole, it is a solid paper with an interesting and novel contribution, but I think it is hampered by the lack of clarity in the areas listed above. I am willing to revise my assessment up if these concerns are addressed.

Questions for the rebuttal period:

Please refer to the questions in the weaknesses section.


---------------------------------------------------------------------------------------------------------------
Update after discussion:
As stated in my initial review, I wanted to see an in depth discussion of what the implications of the binary tree assumption are for the types of transformations that are produced and the impact on the metric using the Jacobian. If these concerns are addressed, I think the paper will be greatly strengthened in the future.

---

> ### Author Response · Authors · 2020-11-23
> **Reply to AnonReviewer3**
>
> We thank the reviewer for the insightful comments, suggestions and questions.
>
>
> (1a) We apologize for the unclarity. A and B indeed refer to two separate time steps. To obtain a time-dependent parameterization, opposed to integration of a single stationary velocity field, we split up the field in a sequence of several time-steps (yielding a piecewise time-dependent parameterization). However, the scaling-and-squaring can only be used to find the exponential of a single field Z and we use the BCH formula to approximate the field Z that corresponds to our time-dependent parameterization. We thank the reviewer for pointing this out and we will clarify this in the final version.
>
> (1b) Now that we have a way to combine two fields A and B into a single field Z, we use a binary tree structure to approximate parameterizations that consist of more than two time steps:
> ```
> Binary tree:                Naive composition:
> ((A, B), (C, D))            ( ( ( (A, B), C), D)
>      /      \                       / \
>    /  \    /  \                   / \   D
>   A    B  C    D                / \  C
>                                A   B
> Max-depth scales Big-O(N)      Max-depth scales Big-O(log N)
> ```
>
> Doing so in a binary tree structure maximally limits the amount of times that the BCH formula, and therefore possible propagation of approximation errors, for each field to Big-O(log N) compared to Big-O(N) if fields are composed sequentially. We added an example composition with N=4 in ascii art above, and will add a more in-depth discussion on this in the ‘Binary Tree Composition’ paragraph on page 4.
>
> (2) All presented measurements, including percentage of negative Jacobian determinant counts, were calculated on entire volumes on a hold-out test set. The TPS models generate inherently smooth fields, and are therefore less prone to folding resulting in fewer negative Jacobian determinants. On the other hand, coarser TPS grids have less flexibility, which would make them unsuitable for application in complex anatomical segmentation tasks. We can see this effect in the application to the breast segmentation task, where the benefit of our method becomes apparent: the diffeomorphic vector fields offer flexible transformations, while respecting essential properties such as preservation of topology, which is supported by the low Hausdorff distances and low number of connected components. We will clarify these considerations in Section 5.2, Section 5.3 and the discussion section.
>
> Thank you for noticing the discrepancy in standard deviations. We are very sorry but see that we have missed to include the proper values for a few standard deviations and will correct this in the manuscript:
>
> U-Net (direct estimation) | | | 0.846 ± 0.24 | 12.62 ± 15.48 | - | 51.79 | \
> Spatial Template Transformer ((Lee et al., 2019)) + Shape Prior (ours) | X | | 0.877 ± 0.21 | 12.58 ± 12.91 | 0.43 ± 0.01 | 19.68 |

---

### Official Review · AnonReviewer1 · 2020-10-28
**Interesting novelty but subtle and not backed up with experimental results**

**Rating:** 3
**Confidence:** 5

**Review:**

Authors present a spatial transformer layer that models
diffeomorphisms. Modelling diffeormophisms with neural networks is not very
new. Prior work successfully utilized stationary vector field prediction and
fast solvers through scaling-and-squaring techniques. These prior work have been
used to tackle registration and segmentation problems. The technical novelty
here is to use time-dependent vector fields for modeling
diffeomorphisms. Experiments on MNIST classification, using the proposed spatial
transformer as a layer, and breast tissue segmentation, using the proposed
transformer as a single network to deform a template, are presented and compared
with conventional spatial transformer layers using thin-plate-splines as
transformation models.

While the introduction of time-dependency is interesting, more from a technical
perspective than a conceptual perspective, I believe there are several aspects
of the paper that needs improvement.

1. Since the main innovation is integration of time-dependency in modelling
vector fields with neural networks, I suggest directly comparing time-dependent
with stationary vector field approaches.
a. Experimental comparison can focus on the differences between stationary and
piece-wise stationary vector field modelling. I could not see this comparison in
the experiments, only a single mention in the conclusion stating better
performance of the proposed model.
b. Analysis of computational time would be interesting. Predicting and
integrating a stationary vector field will be faster I assume. But by how much?

2. MNIST classification experiments raise some concerns:
a. The increase in classification accuracy of the proposed model compared to
CNN + Field-STN is small. This difference may as well be due to randomness of
the optimization. This result is not motivating for the proposed
method. Achieving lower number of negative Jacobian determinants is interesting
but its value is questionable.
b. What is CNN + Field-STN? This is not defined.

3. Segmentation results raise the following concerns:
a. The method proposed here does not seem to be better than the method
proposed in Lee et al. 2019. The lower HD distance can be motivating but value
of this may be better justified. If the proposed innovation was substantial,
lower accuracy would have been completely fine. However, since the innovation is
subtle, such a lower accuracy lowers the enthusiasm for the paper.
b. The proposed method achieves lower number of connected components but the
value of this is not very well motivated from an application
perspective. Providing such a motivation would be helpful.

4. I suggest editing section 3 to present the proposed method much more
clearly. The method only becomes clear in Section 4.

---

> ### Author Response · Authors · 2020-11-19
> **Reply to AnonReviewer1**
>
> We thank the reviewer for the insightful comments, suggestions and questions.
>
> (1) The main novelties of our paper are: (I) using diffeomorphisms to enforce properties on the output of a neural segmentation model, (II) a time-dependent parameterization of the diffeomorphic field using the BCH formula and (III) the analytic shape prior itself (see also answer (3)).
> From a conceptual perspective, using diffeomorphisms (in combination with a carefully chosen shape prior) to enforce certain properties on the output of a neural segmentation model has not been done before and arguably interesting from a conceptual perspective.
>
> (1a) We thank you for this suggestion. The intuition to use a time dependent field is that parameterising the deformation as a time-dependent sequence of velocities allows the network to sequentially model larger movements first and detailed refinements thereafter. In our experiments we did find that the time-dependent parameterisation performed better, but did not run an in-depth analysis of performance between the two and therefore did not include these results in the paper. Nevertheless, we will try to perform additional experiments using stationary fields and provide the results as soon as they are available.
>
> (1b) We have analysed the computation time of our approach (see answer (4) to AnonReviewer4) and in addition also ran an experiment timing the difference between the stationary and time-dependent diffeomorphic fields. The time-dependent vector field is indeed slightly slower compared to a stationary field, although still small enough to be neglectible from a practical perspective.
>
> Method                                                                                    | Inference time averaged over 20 full 3d volumes
>
> U-Net                                                                                       | 1.03 s
> U-Net + non-diffeomorphic field                                        | 1.06 s
> U-Net + stationary diffeomorphic field                             | 1.17 s
> U-Net + time-dependent diffeomorphic field (ours)      | 1.19 s
>
> (2a) For the MNIST results, we tried to apply our method on an existing implementation without tuning hyperparameters in favor of our method to make the comparison as fair as possible. We do agree with the reviewer that the improvement is subtle and the experiments on the breast tissue segmentation might be more convincing as this is a real-world task. The MNIST experiments were repeated 20 times with different seeds to average out randomness in the optimization.
>
> (2b) We apologize for lack of clarity and thank the reviewer for pointing this out. We will update Table 2 and Section 5.2 and clarify that CNN + Field-STN is a spatial transformer network without performing integration of the velocity field to obtain a diffeomorphic field (ie. directly mapping the template with the field outputted by the neural network).
>
> (3a) The method of Lee et al. could not be applied to our task directly, because it relies on a task-specific shape template, which was unavailable for breast segmentation. We extended the method by Lee et al. with our analytical shape prior to act as a template and directly compared the extended approach with our other innovations. We understand this can be confusing and will update Section 5 to clarify this.
> From our results, we did find that integrating the velocity fields generated by the model resulted in slightly worse performance in terms of Dice score, but did clearly outperform the other methods in terms of Hausdorff Distance (HD), number of negative Jacobian determinants (<%JD) and number of connected components (CC). Also, we will update the importance of CC and HD for this task in Section 5 of the manuscript.
>
> (3b) In medical imaging tasks, robustness of the output is extremely important. We know beforehand that breast tissue is a single connected component and the edges should be smooth. By carefully choosing the shape prior to obey these properties, we enforce these properties - by definition of continuity of the diffeomorphic transformation - also to the output of the network. We are aware that we have missed some of these clinical motivations of the work and will add them in Section 1 and Section 5 of the manuscript.
>
> (4) Thank you for the advice. We will rewrite as suggested and make sure the template transformer is clearly introduced earlier in the paper.

---

### Official Review · AnonReviewer2 · 2020-10-30
**Official Blind Review #2**

**Rating:** 6
**Confidence:** 2

**Review:**

This paper propose a novel method to incorporate shape prior in neural networks based on Diffeomorphic transformation. This is useful as by design it preserves certain desirable properties of output such as smooth boundaries and connected components which are of interest in medical imaging applications.   The method is validated on Mnist for data invariance and a medical imaging task for segmentation.

Novelty: The idea of  incorporating shape prior information into neural network based image segmentation is inspired by Lee et al. This method shows  how to use a diffeomorphic spatial transformer to warp a shape prior where warping is based on time-dependent parameterisation of multiple vector fields utilizing the Baker-Campbell-Hausdorff formula.

Clarity and Presentation: The paper is overall well written and motivated. Minor comment: As an outsider, I found the argument on negative Jacobian determinants hard to follow and it comes up several times. So it seems quite important.

Experimental Validation: It is partially on weaker side in my opinion.  e.g. Lee et al. experiments were shown on segmentation of coronary lumen structures. Is there a good reason to instead choose breast tissue segmentation task only and not show any experiments on the former one? similarly, to prove data invariance, STN is validated on several benchmarks whereas this method only uses one, MNIST.

---

> ### Author Response · Authors · 2020-11-19
> **Reply to AnonReviewer2**
>
> We thank the reviewer for suggestions and questions.
>
>
> Clarity and Presentation:
>
> The ratio of negative Jacobian determinants is a standard method to evaluate presence of undesired image folding in deformation fields. Although we found that integrating the velocity fields generated by the network helped to reduce the number of negative Jacobians (see Table 2 on page 7), they still occur. Further reduction of these negative Jacobians (i.e. foldings) is an interesting open research question.
>
> Experimental Validation:
>
> Breast tissue segmentation is a particularly interesting problem for this task, because we know that the breast tissue is always a single connected component. Since we propose our diffeomorphic template transformer to enforce this property on the output of a segmentation model, it is a suitable task to evaluate performance. It would be very interesting to assess how our proposed method would perform on other tasks such as coronary artery tree segmentation, however we are not able to use the data used by Lee et. al since that data is not publicly available. We will try to better this in Section 6 on page 8.
>
> Regarding the experiments showing data invariances, we were able to perform better on an existing implementation without any tuning (please see our answer to AnonReviewer4-3), which shows the general applicability of diffeomorphic spatial transformers. It would indeed be interesting to see how well the approach can learn data invariances on other tasks, but chose to put more emphasis on the breast tissue experiments since diffeomorphisms lend themselves particularly to template transformer setting, to enforce properties on the output.

---

### Official Review · AnonReviewer4 · 2020-11-01
**Incremental methodological contribution and performance improvement**

**Rating:** 5
**Confidence:** 4

**Review:**

This submission proposes a diffeomorphic spatial transform network, which considers the specific transformation, i.e., diffeomorphism, in the data. The research topic is quite interesting, but the submission is at the preliminary stage and needs more work before publishing.

1) The writing of the paper could be improved in terms of clarification. Most of the time, the authors present the solutions without explaining the motivations or underlying intuitions. For instance, why do we have to have the time-dependent vector field? Why do we need to use the prior shape?

2) The novelty of the proposed method is limited. The main contribution of this paper is the use of BCH formula to approximate the piece-wise time-dependent sequence of the vector field. Very likely, this approximation is not very efficient, and we also don't know its approximation accuracy.

3) The performance improvement is limited. In the MNIST experimental results, the accuracy improvement is subtle. Compared to CNN+Field-STN, the CNN+Diffeomorhpic-STN has a slightly increased mean accuracy with a larger standard deviation and a slightly decreased number of negative Jacobians with also a larger standard deviation. For the breast tissue segmentation task, if we consider the transformer only (without considering the shape prior), the Dice Score performance of the proposed method is downgraded.

4) The efficiency of the proposed method is questionable. If we use this diffeomorphic transform network as a plugin module in another network, will it become the computational bottleneck of the whole model, since the estimated spatiotemporal vector field is really high-dimensional?

5) Minor question: Is the identity map missing in the Algirhtm 1? phi_0 = Id + v/2^T?

---

> ### Author Response · Authors · 2020-11-19
> **Reply to AnonReviewer4**
>
> We thank the reviewer for the insightful comments, suggestions and questions.
>
> (1) We apologize for the unclarity. We will clarify the motivations and intuitions behind the time-dependent vector field and the prior shape in Section 3 and Section 4.1 of the manuscript.
>
> The main intuition about parameterizing a time-dependent vector field is increased flexibility. The network needs to deform a ball shaped object into the shape of the breast tissue and then obtain a fine detailed alignment of the shape around the edges of the shape. Parameterising the deformation as a time-dependent sequence of velocities allows the network to sequentially model larger movements first and detailed refinements thereafter.
>
> The main intuition behind the prior shape is that it is required to construct an object such that a single connected component and smooth boundaries are guaranteed. By definition of continuity of the diffeomorphic transformation, this will also enforce these properties on the model output. In this paper we propose an analytic function that defines a gaussian-like ball shape to be used as a template. The motivation behind our proposed shape is twofold: First, we show that our method is flexible enough to deform even simple ball-shape functions into highly accurate output predictions of breast tissue. Second, the Gaussian ball is generic, making the prior shape practical from an engineering perspective. Nevertheless, any template that fits the specific problem at hand can be implemented.
>
> (2) The main novelties of our paper are: (I) using diffeomorphisms to enforce desirable properties on the output of a neural segmentation model, (II) an efficient time-dependent parameterization of the diffeomorphic field using the BCH formula and (III) the proposed analytic shape prior itself.
> Like any other approximation method, our approximation based on the BCH formula has some errors, but the results corroborate that the effect on model performance is likely limited. The benefit of using the BCH formula is that it allows us to use scaling-and-squaring to approximate a time-dependent field making it very efficient (please see answer (4) below). We will add a comparison of computational time measurements supporting this in the Supplementary Material.
>
> (3) We agree with the reviewer that compared to conventional spatial transformer networks the improvements on MNIST are subtle, but we did achieve a performance increase nevertheless. However, we would like to emphasize that for the comparison we used settings that were specifically designed for the conventional methods, and we did not tune any hyperparameters in favor of our proposed method.
> We agree with the reviewer that our method did not perform best in terms of Dice score, but it outperformed the other methods in all other metrics. In medical tasks, robustness and consistency of methods is extremely important, since here downstream tasks may depend on correct continuous segmentation. These criteria are better reflected by including the other evaluation metrics: negative Jacobian determinants, Hausdorff Distance (HD), and connected components (CC). Moreover, close visual inspection of the results revealed that the outputs are qualitatively better.
>
> (4) We specifically have proposed to use the BCH formula to be able to approximate the time-dependent parameterisation using scaling-and-squaring, which is highly parallelizable and one of the most efficient approximations to calculate the matrix exponential. We will update our manuscript and add a comparison of computational time supporting this in the Supplementary Material:
>
> Method                                                   | Inference time averaged over 20 full 3d volumes
>
> U-Net                                                       | 1.03 s
> U-Net + non-diffeomorphic field        | 1.06 s
> U-Net + diffeomorphic field (ours)     | 1.19 s
>
> (5) Thank you for pointing this out. The scaling-and-squaring algorithm is very similar to Euler’s method, where you would start from the identity Id and exponentiate by iteratively applying the field scaled down by the time fraction. In scaling-and-squaring, however, you start by ‘scaling’ down the field (v/2^T) and then self-composing (‘squaring’) in order to find the solution more efficiently (see Moler and Van Loan, 2003). We understand how this can be confusing and will explain the difference between the two more clearly in the final version.

---

### Decision · Program_Chairs · 2021-01-07
**Final Decision**

**Decision:**

Reject

**Comment:**

This paper has initially received mixed reviews, with two favorable and two unfavorable reviews. Several serious issues have been raised, in particular on experiments and validation; limited novelty; limited performance improvement; on the preliminary stage of the paper, in particular presentation and writing, and on the justification of key choices.

The authors provided responses to some of these issues, but in the discussion phase the reviewers (and the AC) judged the the responses did not sufficiently address the weaknesses of the paper, in particular:
- The experimental setup does not assess the key innovation of the paper.
- Several contributions claimed by the authors (in the paper and in the response) are judged not to be novel.
- Concerns regarding the metric chosen to measure smoothness.
and other issues.

The reviewers and AC agreed, that the paper has potential and merits, but that at at this point it is not yet ready for publication.